# Measuring Neural Net Robustness with Constraints

**Osbert Bastani**
Stanford University
obastani@cs.stanford.edu

**Yani Ioannou**
University of Cambridge
yai20@cam.ac.uk

**Leonidas Lampropoulos**
University of Pennsylvania
llamp@seas.upenn.edu

**Dimitrios Vytiniotis**
Microsoft Research
dimitris@microsoft.com

**Aditya V. Nori**
Microsoft Research
adityan@microsoft.com

**Antonio Criminisi**
Microsoft Research
antcrim@microsoft.com

## Abstract

Despite having high accuracy, neural nets have been shown to be susceptible to adversarial examples, where a small perturbation to an input can cause it to become mislabeled. We propose metrics for measuring the robustness of a neural net and devise a novel algorithm for approximating these metrics based on an encoding of robustness as a linear program. We show how our metrics can be used to evaluate the robustness of deep neural nets with experiments on the MNIST and CIFAR-10 datasets. Our algorithm generates more informative estimates of robustness metrics compared to estimates based on existing algorithms. Furthermore, we show how existing approaches to improving robustness "overfit" to adversarial examples generated using a specific algorithm. Finally, we show that our techniques can be used to additionally improve neural net robustness both according to the metrics that we propose, but also according to previously proposed metrics.

## 1 Introduction

Recent work [21] shows that it is often possible to construct an input mislabeled by a neural net by perturbing a correctly labeled input by a tiny amount in a carefully chosen direction. Lack of robustness can be problematic in a variety of settings, such as changing camera lens or lighting conditions, successive frames in a video, or adversarial attacks in security-critical applications [18].

A number of approaches have since been proposed to improve robustness [6, 5, 1, 7, 20]. However, work in this direction has been handicapped by the lack of objective measures of robustness. A typical approach to improving the robustness of a neural net $f$ is to use an algorithm $\mathcal{A}$ to find adversarial examples, augment the training set with these examples, and train a new neural net $f'$ [5]. Robustness is then evaluated by using the *same* algorithm $\mathcal{A}$ to find adversarial examples for $f'$—if $\mathcal{A}$ discovers fewer adversarial examples for $f'$ than for $f$, then $f'$ is concluded to be more robust than $f$. However, $f'$ may have *overfit* to adversarial examples generated by $\mathcal{A}$—in particular, a *different* algorithm $\mathcal{A}'$ may find as many adversarial examples for $f'$ as for $f$. Having an objective robustness measure is vital not only to reliably compare different algorithms, but also to understand robustness of production neural nets—e.g., when deploying a login system based on face recognition, a security team may need to evaluate the risk of an attack using adversarial examples.

In this paper, we study the problem of measuring robustness. We propose to use two statistics of the robustness $\rho(f, \mathbf{x}_*)$ of $f$ at point $\mathbf{x}_*$ (i.e., the $L_\infty$ distance from $\mathbf{x}_*$ to the nearest adversarial example) [21]. The first one measures the frequency with which adversarial examples occur; the other measures the severity of such adversarial examples. Both statistics depend on a parameter $\epsilon$, which intuitively specifies the threshold below which adversarial examples should not exist (i.e., points $\mathbf{x}$ with $L_\infty$ distance to $\mathbf{x}_*$ less than $\epsilon$ should be assigned the same label as $\mathbf{x}_*$).

The key challenge is efficiently computing $\rho(f, \mathbf{x}_*)$. We give an exact formulation of this problem as an intractable optimization problem. To recover tractability, we approximate this optimization problem by constraining the search to a convex region $\mathcal{Z}(\mathbf{x}_*)$ around $\mathbf{x}_*$. Furthermore, we devise an iterative approach to solving the resulting linear program that produces an order of magnitude speed-up. Common neural nets (specifically, those using rectified linear units as activation functions) are in fact piecewise linear functions [15]; we choose $\mathcal{Z}(\mathbf{x}_*)$ to be the region around $\mathbf{x}_*$ on which $f$ is linear. Since the linear nature of neural nets is often the cause of adversarial examples [5], our choice of $\mathcal{Z}(\mathbf{x}_*)$ focuses the search where adversarial examples are most likely to exist.

We evaluate our approach on a deep convolutional neural network $f$ for MNIST. We estimate $\rho(f, \mathbf{x}_*)$ using both our algorithm $\mathcal{A}_{\text{LP}}$ and (as a baseline) the algorithm $\mathcal{A}_{\text{L-BFGS}}$ introduced by [21]. We show that $\mathcal{A}_{\text{LP}}$ produces a substantially more accurate estimate of $\rho(f, \mathbf{x}_*)$ than $\mathcal{A}_{\text{L-BFGS}}$. We then use data augmentation with each algorithm to improve the robustness of $f$, resulting in *fine-tuned* neural nets $f_{\text{LP}}$ and $f_{\text{L-BFGS}}$. According to $\mathcal{A}_{\text{L-BFGS}}$, $f_{\text{L-BFGS}}$ is more robust than $f$, but *not* according to $\mathcal{A}_{\text{LP}}$. In other words, $f_{\text{L-BFGS}}$ overfits to adversarial examples computed using $\mathcal{A}_{\text{L-BFGS}}$. In contrast, $f_{\text{LP}}$ is more robust according to both $\mathcal{A}_{\text{L-BFGS}}$ and $\mathcal{A}_{\text{LP}}$. Furthermore, to demonstrate scalability, we apply our approach to evaluate the robustness of the 23-layer network-in-network (NiN) neural net [13] for CIFAR-10, and reveal a surprising lack of robustness. We fine-tune NiN and show that robustness improves, albeit only by a small amount. In summary, our contributions are:

- We formalize the notion of pointwise robustness studied in previous work [5, 21, 6] and propose two statistics for measuring robustness based on this notion (§2).
- We show how computing pointwise robustness can be encoded as a constraint system (§3). We approximate this constraint system with a tractable linear program and devise an optimization for solving this linear program an order of magnitude faster (§4).
- We demonstrate experimentally that our algorithm produces substantially more accurate measures of robustness compared to algorithms based on previous work, and show evidence that neural nets fine-tuned to improve robustness (§5) can overfit to adversarial examples identified by a specific algorithm (§6).

## 1.1   Related work

The susceptibility of neural nets to adversarial examples was discovered by [21]. Given a test point $\mathbf{x}_*$ with predicted label $\ell_*$, an adversarial example is an input $\mathbf{x}_* + \mathbf{r}$ with predicted label $\ell \neq \ell_*$ where the adversarial perturbation $\mathbf{r}$ is small (in $L_\infty$ norm). Then, [21] devises an approximate algorithm for finding the smallest possible adversarial perturbation $\mathbf{r}$. Their approach is to minimize the combined objective $\text{loss}(f(\mathbf{x}_* + \mathbf{r}), \ell) + c\|\mathbf{r}\|_\infty$, which is an instance of box-constrained convex optimization that can be solved using L-BFGS-B. The constant $c$ is optimized using line search.

Our formalization of the robustness $\rho(f, \mathbf{x}_*)$ of $f$ at $\mathbf{x}_*$ corresponds to the notion in [21] of finding the minimal $\|\mathbf{r}\|_\infty$. We propose an exact algorithm for computing $\rho(f, \mathbf{x}_*)$ as well as a tractable approximation. The algorithm in [21] can also be used to approximate $\rho(f, \mathbf{x}_*)$; we show experimentally that our algorithm is substantially more accurate than [21].

There has been a range of subsequent work studying robustness; [17] devises an algorithm for finding purely synthetic adversarial examples (i.e., no initial image $\mathbf{x}_*$), [22] searches for adversarial examples using random perturbations, showing that adversarial examples in fact exist in large regions of the pixel space, [19] shows that even intermediate layers of neural nets are not robust to adversarial noise, and [3] seeks to explain why neural nets may generalize well despite poor robustness properties.

Starting with [5], a major focus has been on devising faster algorithms for finding adversarial examples. Their idea is that adversarial examples can then be computed on-the-fly and used as training examples, analogous to data augmentation approaches typically used to train neural nets [10]. To find adversarial examples quickly, [5] chooses the adversarial perturbation $\mathbf{r}$ to be in the direction of the signed gradient of $\text{loss}(f(\mathbf{x}_* + \mathbf{r}), \ell)$ with fixed magnitude. Intuitively, given only the gradient of the loss function, this choice of $\mathbf{r}$ is most likely to produce an adversarial example with $\|r\|_\infty \leq \epsilon$. In this direction, [16] improves upon [5] by taking multiple gradient steps, [7] extends this idea to norms beyond the $L_\infty$ norm, [6] takes the approach of [21] but fixes $c$, and [20] formalizes [5] as robust optimization.

A key shortcoming of these lines of work is that robustness is typically measured using the same algorithm used to find adversarial examples, in which case the resulting neural net may have overfit

to adversarial examples generating using that algorithm. For example, [5] shows improved accuracy to adversarial examples generated using their own signed gradient method, but do not consider whether robustness increases for adversarial examples generated using more precise approaches such as [21]. Similarly, [7] compares accuracy to adversarial examples generated using both itself and [5] (but not [21]), and [20] only considers accuracy on adversarial examples generated using their own approach on the baseline network. The aim of our paper is to provide metrics for evaluating robustness, and to demonstrate the importance of using such impartial measures to compare robustness.

Additionally, there has been work on designing neural network architectures [6] and learning procedures [1] that improve robustness to adversarial perturbations, though they do not obtain state-of-the-art accuracy on the unperturbed test sets. There has also been work using smoothness regularization related to [5] to train neural nets, focusing on improving accuracy rather than robustness [14].

Robustness has also been studied in more general contexts; [23] studies the connection between robustness and generalization, [2] establishes theoretical lower bounds on the robustness of linear and quadratic classifiers, and [4] seeks to improve robustness by promoting resiliance to deleting features during training. More broadly, robustness has been identified as a desirable property of classifiers beyond prediction accuracy. Traditional metrics such as (out-of-sample) accuracy, precision, and recall help users assess prediction accuracy of trained models; our work aims to develop analogous metrics for assessing robustness.

## 2  Robustness Metrics

Consider a classifier $f : \mathcal{X} \to \mathcal{L}$, where $\mathcal{X} \subseteq \mathbb{R}^n$ is the input space and $\mathcal{L} = \{1, ..., L\}$ are the labels. We assume that training and test points $\mathbf{x} \in \mathcal{X}$ have distribution $\mathcal{D}$. We first formalize the notion of robustness at a point, and then describe two statistics to measure robustness. Our two statistics depend on a parameter $\epsilon$, which captures the idea that we only care about robustness below a certain threshold—we disregard adversarial examples $\mathbf{x}$ whose $L_\infty$ distance to $\mathbf{x}_*$ is greater than $\epsilon$. We use $\epsilon = 20$ in our experiments on MNIST and CIFAR-10 (on the pixel scale 0-255).

**Pointwise robustness.** Intuitively, $f$ is robust at $\mathbf{x}_* \in \mathcal{X}$ if a "small" perturbation to $\mathbf{x}_*$ does not affect the assigned label. We are interested in perturbations sufficiently small that they do not affect human classification; an established condition is $\|\mathbf{x} - \mathbf{x}_*\|_\infty \leq \epsilon$ for some parameter $\epsilon$. Formally, we say $f$ is $(\mathbf{x}_*, \epsilon)$-*robust* if for every $\mathbf{x}$ such that $\|\mathbf{x} - \mathbf{x}_*\|_\infty \leq \epsilon$, $f(\mathbf{x}) = f(\mathbf{x}_*)$. Finally, the *pointwise robustness* $\rho(f, \mathbf{x}_*)$ of $f$ at $\mathbf{x}_*$ is the minimum $\epsilon$ for which $f$ fails to be $(\mathbf{x}_*, \epsilon)$-robust:

$$\rho(f, \mathbf{x}_*) \stackrel{\text{def}}{=} \inf\{\epsilon \geq 0 \mid f \text{ is not } (\mathbf{x}_*, \epsilon)\text{-robust}\}. \tag{1}$$

This definition formalizes the notion of robustness in [5, 6, 21].

**Adversarial frequency.**  Given a parameter $\epsilon$, the *adversarial frequency*

$$\phi(f, \epsilon) \stackrel{\text{def}}{=} \Pr_{\mathbf{x}_* \sim \mathcal{D}}[\rho(f, \mathbf{x}_*) \leq \epsilon]$$

measures how often $f$ fails to be $(\mathbf{x}_*, \epsilon)$-robust. In other words, if $f$ has high adversarial frequency, then it fails to be $(\mathbf{x}_*, \epsilon)$-robust for many inputs $\mathbf{x}_*$.

**Adversarial severity.**  Given a parameter $\epsilon$, the *adversarial severity*

$$\mu(f, \epsilon) \stackrel{\text{def}}{=} \mathbb{E}_{\mathbf{x}_* \sim \mathcal{D}}[\rho(f, \mathbf{x}_*) \mid \rho(f, \mathbf{x}_*) \leq \epsilon]$$

measures the severity with which $f$ fails to be robust at $\mathbf{x}_*$ conditioned on $f$ not being $(\mathbf{x}_*, \epsilon)$-robust. We condition on pointwise robustness since once $f$ is $(\mathbf{x}_*, \epsilon)$-robust at $\mathbf{x}_*$, then the degree to which $f$ is robust at $\mathbf{x}_*$ does not matter. *Smaller $\mu(f, \epsilon)$ corresponds to worse* adversarial severity, since $f$ is more susceptible to adversarial examples if the distances to the nearest adversarial example are small.

The frequency and severity capture different robustness behaviors. A neural net may have high adversarial frequency but low adversarial severity, indicating that most adversarial examples are about $\epsilon$ distance away from the original point $\mathbf{x}_*$. Conversely, a neural net may have low adversarial frequency but high adversarial severity, indicating that it is typically robust, but occasionally severely fails to be robust. Frequency is typically the more important metric, since a neural net with low adversarial frequency is robust most of the time. Indeed, adversarial frequency corresponds to the

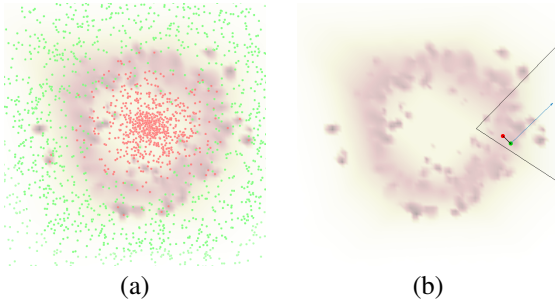

(a)          (b)

Figure 1: Neural net with a single hidden layer and ReLU activations trained on dataset with binary labels. (a) The training data and loss surface. (b) The linear region corresponding to the red training point.

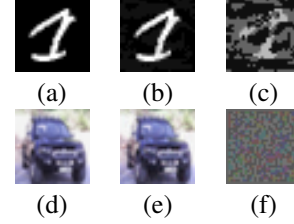

(a)     (b)     (c)

(d)     (e)     (f)

Figure 2: For MNIST, (a) an image classified 1, (b) its adversarial example classifed 3, and (c) the (scaled) adversarial perturbation. For CIFAR-10, (d) an image classified as "automobile", (e) its adversarial example classified as "truck", and (f) the (scaled) adversarial perturbation.

accuracy on adversarial examples used to measure robustness in [5, 20]. Severity can be used to differentiate between neural nets with similar adversarial frequency.

Given a set of samples $X \subseteq \mathcal{X}$ drawn i.i.d. from $\mathcal{D}$, we can estimate $\phi(f, \epsilon)$ and $\mu(f, \epsilon)$ using the following standard estimators, assuming we can compute $\rho$:

$$\hat{\phi}(f, \epsilon, X) \stackrel{\text{def}}{=} \frac{|\{\mathbf{x}_* \in X \mid \rho(f, \mathbf{x}_*) \leq \epsilon\}|}{|X|}$$

$$\hat{\mu}(f, \epsilon, X) \stackrel{\text{def}}{=} \frac{\sum_{\mathbf{x}_* \in X} \rho(f, \mathbf{x}_*) \mathbb{I}[\rho(f, \mathbf{x}_*) \leq \epsilon]}{|\{\mathbf{x}_* \in X \mid \rho(f, \mathbf{x}_*) \leq \epsilon\}|}.$$

An approximation $\hat{\rho}(f, \mathbf{x}_*) \approx \rho(f, \mathbf{x}_*)$ of $\rho$, such as the one we describe in Section 4, can be used in place of $\rho$. In practice, $X$ is taken to be the test set $X_{\text{test}}$.

## 3 Computing Pointwise Robustness

### 3.1 Overview

Consider the training points in Figure 1 (a) colored based on the ground truth label. To classify this data, we train a two-layer neural net $f(x) = \arg\max_\ell \{(W_2 g(W_1 \mathbf{x}))_\ell\}$, where the ReLU function $g$ is applied pointwise. Figure 1 (a) includes contours of the per-point loss function of this neural net.

Exhaustively searching the input space to determine the distance $\rho(f, \mathbf{x}_*)$ to the nearest adversarial example for input $\mathbf{x}_*$ (labeled $\ell_*$) is intractable. Recall that neural nets with rectified-linear (ReLU) units as activations are piecewise linear [15]. Since adversarial examples exist because of this linearity in the neural net [5], we restrict our search to the region $\mathcal{Z}(\mathbf{x}_*)$ around $\mathbf{x}_*$ on which the neural net is linear. This region around $\mathbf{x}_*$ is defined by the activation of the ReLU function: for each $i$, if $(W_1 \mathbf{x}_*)_i \geq 0$ (resp., $(W_1 \mathbf{x}_*) \leq 0$), we constrain to the half-space $\{\mathbf{x} \mid (W_1 \mathbf{x})_i \geq 0\}$ (resp., $\{\mathbf{x} \mid (W_1 \mathbf{x})_i \leq 0\}$). The intersection of these half-spaces is convex, so it admits efficient search. Figure 1 (b) shows one such convex region [1].

Additionally, $\mathbf{x}$ is labeled $\ell$ exactly when $f(\mathbf{x})_\ell \geq f(\mathbf{x})_{\ell'}$ for each $\ell' \neq \ell$. These constraints are linear since $f$ is linear on $\mathcal{Z}(\mathbf{x}_*)$. Therefore, we can find the distance to the nearest input with label $\ell \neq \ell_*$ by minimizing $\|\mathbf{x} - \mathbf{x}_*\|_\infty$ on $\mathcal{Z}(\mathbf{x}_*)$. Finally, we can perform this search for each label $\ell \neq \ell_*$, though for efficiency we take $\ell$ to be the label assigned the second-highest score by $f$. Figure 1 (b) shows the adversarial example found by our algorithm in our running example. In Figure 1 note that the direction of the nearest adversarial example is not necessary aligned with the signed gradient of the loss function, as observed by others [7].

## 3.2 Formulation as Optimization

We compute $\rho(f, \epsilon)$ by expressing (1) as *constraints* $\mathcal{C}$, which consist of

- Linear relations; specifically, inequalities $\mathcal{C} \equiv (\mathbf{w}^T \mathbf{x} + b \geq 0)$ and equalities $\mathcal{C} \equiv (\mathbf{w}^T \mathbf{x} + b = 0)$, where $\mathbf{x} \in \mathbb{R}^m$ (for some $m$) are variables and $\mathbf{w} \in \mathbb{R}^m$, $b \in \mathbb{R}$ are constants.
- Conjunctions $\mathcal{C} \equiv \mathcal{C}_1 \wedge \mathcal{C}_2$, where $\mathcal{C}_1$ and $\mathcal{C}_2$ are themselves constraints. Both constraints must be satisfied for the conjunction to be satisfied.
- Disjunctions $\mathcal{C} \equiv \mathcal{C}_1 \vee \mathcal{C}_2$, where $\mathcal{C}_1$ and $\mathcal{C}_2$ are themselves constraints. One of the constraints must be satisfied for the disjunction to be satisfied.

The *feasible set* $\mathcal{F}(\mathcal{C})$ of $\mathcal{C}$ is the set of $\mathbf{x} \in \mathbb{R}^m$ that satisfy $\mathcal{C}$; $\mathcal{C}$ is *satisfiable* if $\mathcal{F}(\mathcal{C})$ is nonempty.

In the next section, we show that the condition $f(\mathbf{x}) = \ell$ can be expressed as constraints $\mathcal{C}_f(\mathbf{x}, \ell)$; i.e., $f(\mathbf{x}) = \ell$ if and only if $\mathcal{C}_f(\mathbf{x}, \ell)$ is satisfiable. Then, $\rho(f, \epsilon)$ can be computed as follows:

$$\rho(f, \mathbf{x}_*) = \min_{\ell \neq \ell_*} \rho(f, \mathbf{x}_*, \ell) \tag{2}$$

$$\rho(f, \mathbf{x}_*, \ell) \stackrel{\text{def}}{=} \inf\{\epsilon \geq 0 \mid \mathcal{C}_f(\mathbf{x}, \ell) \wedge \|\mathbf{x} - \mathbf{x}_*\|_\infty \leq \epsilon \text{ satisfiable}\}. \tag{3}$$

The optimization problem is typically intractable; we describe a tractable approximation in §4.

## 3.3 Encoding a Neural Network

We show how to encode the constraint $f(\mathbf{x}) = \ell$ as constraints $\mathcal{C}_f(\mathbf{x}, \ell)$ when $f$ is a neural net. We assume $f$ has form $f(\mathbf{x}) = \arg\max_{\ell \in \mathcal{L}} \left\{ \left[ f^{(k)}(f^{(k-1)}(...(f^{(1)}(\mathbf{x}))...)) \right]_\ell \right\}$, where the $i^{th}$ layer of the network is a function $f^{(i)} : \mathbb{R}^{n_{i-1}} \to \mathbb{R}^{n_i}$, with $n_0 = n$ and $n_k = |\mathcal{L}|$. We describe the encoding of fully-connected and ReLU layers; convolutional layers are encoded similarly to fully-connected layers and max-pooling layers are encoded similarly to ReLU layers. We introduce the variables $\mathbf{x}^{(0)}, \dots, \mathbf{x}^{(k)}$ into our constraints, with the interpretation that $\mathbf{x}^{(i)}$ represents the output vector of layer $i$ of the network; i.e., $\mathbf{x}^{(i)} = f^{(i)}(\mathbf{x}^{(i-1)})$. The constraint $\mathcal{C}_{\text{in}}(\mathbf{x}) \equiv (\mathbf{x}^{(0)} = \mathbf{x})$ encodes the input layer. For each layer $f^{(i)}$, we encode the computation of $\mathbf{x}^{(i)}$ given $\mathbf{x}^{(i-1)}$ as a constraint $\mathcal{C}_i$.

**Fully-connected layer.** In this case, $\mathbf{x}^{(i)} = f^{(i)}(\mathbf{x}^{(i-1)}) = W^{(i)}\mathbf{x}^{(i-1)} + \mathbf{b}^{(i)}$, which we encode using the constraints $\mathcal{C}_i \equiv \bigwedge_{j=1}^{n_i} \left\{ \mathbf{x}_j^{(i)} = W_j^{(i)}\mathbf{x}^{(i-1)} + \mathbf{b}_j^{(i)} \right\}$, where $W_j^{(i)}$ is the $j$-th row of $W^{(i)}$.

**ReLU layer.** In this case, $\mathbf{x}_j^{(i)} = \max \left\{ \mathbf{x}_j^{(i-1)}, 0 \right\}$ (for each $1 \leq j \leq n_i$), which we encode using the constraints $\mathcal{C}_i \equiv \bigwedge_{j=1}^{n_i} \mathcal{C}_{ij}$, where $\mathcal{C}_{ij} = (\mathbf{x}_j^{(i-1)} < 0 \wedge \mathbf{x}_j^{(i)} = 0) \vee (\mathbf{x}_j^{(i-1)} \geq 0 \wedge \mathbf{x}_j^{(i)} = \mathbf{x}_j^{(i-1)})$.

Finally, the constraints $\mathcal{C}_{\text{out}}(\ell) \equiv \bigwedge_{\ell' \neq \ell} \left\{ \mathbf{x}_\ell^{(k)} \geq \mathbf{x}_{\ell'}^{(k)} \right\}$ ensure that the output label is $\ell$. Together, the constraints $\mathcal{C}_f(\mathbf{x}, \ell) \equiv \mathcal{C}_{\text{in}}(\mathbf{x}) \wedge \left( \bigwedge_{i=1}^{k} \mathcal{C}_i \right) \wedge \mathcal{C}_{\text{out}}(\ell)$ encodes the computation of $f$:

**Theorem 1** For any $\mathbf{x} \in \mathcal{X}$ and $\ell \in \mathcal{L}$, we have $f(\mathbf{x}) = \ell$ if and only if $\mathcal{C}_f(\mathbf{x}, \ell)$ is satisfiable.

# 4 Approximate Computation of Pointwise Robustness

**Convex restriction.** The challenge to solving (3) is the non-convexity of the feasible set of $\mathcal{C}_f(\mathbf{x}, \ell)$. To recover tractability, we approximate (3) by constraining the feasible set to $\mathbf{x} \in \mathcal{Z}(\mathbf{x}_*)$, where $\mathcal{Z}(\mathbf{x}_*) \subseteq \mathcal{X}$ is carefully chosen so that the constraints $\hat{\mathcal{C}}_f(\mathbf{x}, \ell) \equiv \mathcal{C}_f(\mathbf{x}, \ell) \wedge (\mathbf{x} \in \mathcal{Z}(\mathbf{x}_*))$ have convex feasible set. We call $\hat{\mathcal{C}}_f(\mathbf{x}, \ell)$ the *convex restriction* of $\mathcal{C}_f(\mathbf{x}, \ell)$. In some sense, convex restriction is the opposite of convex relaxation. Then, we can approximately compute robustness:

$$\hat{\rho}(f, \mathbf{x}_*, \ell) \stackrel{\text{def}}{=} \inf\{\epsilon \geq 0 \mid \hat{\mathcal{C}}_f(\mathbf{x}, \ell) \wedge \|\mathbf{x} - \mathbf{x}_*\|_\infty \leq \epsilon \text{ satisfiable}\}. \tag{4}$$

The objective is optimized over $\mathbf{x} \in \mathcal{Z}(\mathbf{x}_*)$, which approximates the optimum over $\mathbf{x} \in \mathcal{X}$.

**Choice of $\mathcal{Z}(\mathbf{x}_*)$.** We construct $\mathcal{Z}(\mathbf{x}_*)$ as the feasible set of constraints $\mathcal{D}(\mathbf{x}_*)$; i.e., $\mathcal{Z}(\mathbf{x}_*) = \mathcal{F}(\mathcal{D}(\mathbf{x}_*))$. We now describe how to construct $\mathcal{D}(\mathbf{x}_*)$.

Note that $\mathcal{F}(\mathbf{w}^T\mathbf{x} + b = 0)$ and $\mathcal{F}(\mathbf{w}^T\mathbf{x} + b \geq 0)$ are convex sets. Furthermore, if $\mathcal{F}(\mathcal{C}_1)$ and $\mathcal{F}(\mathcal{C}_2)$ are convex, then so is their conjunction $\mathcal{F}(\mathcal{C}_1 \wedge \mathcal{C}_2)$. However, their disjunction $\mathcal{F}(\mathcal{C}_1 \vee \mathcal{C}_2)$ may not be convex; for example, $\mathcal{F}((x \geq 0) \vee (y \geq 0))$. The potential non-convexity of disjunctions makes (3) difficult to optimize.

We can eliminate disjunction operations by choosing one of the two disjuncts to hold. For example, note that for $\mathcal{C}_1 \equiv \mathcal{C}_2 \vee \mathcal{C}_3$, we have both $\mathcal{F}(\mathcal{C}_2) \subseteq \mathcal{F}(\mathcal{C}_1)$ and $\mathcal{F}(\mathcal{C}_3) \subseteq \mathcal{F}(\mathcal{C}_1)$. In other words, if we replace $\mathcal{C}_1$ with either $\mathcal{C}_2$ or $\mathcal{C}_3$, the feasible set of the resulting constraints can only become smaller. Taking $\mathcal{D}(\mathbf{x}_*) \equiv \mathcal{C}_2$ (resp., $\mathcal{D}(\mathbf{x}_*) \equiv \mathcal{C}_3$) effectively replaces $\mathcal{C}_1$ with $\mathcal{C}_2$ (resp., $\mathcal{C}_3$).

To restrict (3), for every disjunction $\mathcal{C}_1 \equiv \mathcal{C}_2 \vee \mathcal{C}_3$, we systematically choose either $\mathcal{C}_2$ or $\mathcal{C}_3$ to replace the constraint $\mathcal{C}_1$. In particular, we choose $\mathcal{C}_2$ if $\mathbf{x}_*$ satisfies $\mathcal{C}_2$ (i.e., $\mathbf{x}_* \in \mathcal{F}(\mathcal{C}_2)$) and choose $\mathcal{C}_3$ otherwise. In our constraints, disjunctions are always mutually exclusive, so $\mathbf{x}_*$ never simultaneously satisfies both $\mathcal{C}_2$ and $\mathcal{C}_3$. We then take $\mathcal{D}(\mathbf{x}_*)$ to be the conjunction of all our choices. The resulting constraints $\hat{\mathcal{C}}_f(\mathbf{x}, \ell)$ contains only conjunctions of linear relations, so its feasible set is convex. In fact, it can be expressed as a linear program (LP) and can be solved using any standard LP solver.

For example, consider a rectified linear layer (as before, max pooling layers are similar). The original constraint added for unit $j$ of rectified linear layer $f^{(i)}$ is

$$\left(\mathbf{x}_j^{(i-1)} \leq 0 \wedge \mathbf{x}_j^{(i)} = 0\right) \vee \left(\mathbf{x}_j^{(i-1)} \geq 0 \wedge \mathbf{x}_j^{(i)} = \mathbf{x}_j^{(i-1)}\right)$$

To restrict this constraint, we evaluate the neural network on the seed input $\mathbf{x}_*$ and look at the input to $f^{(i)}$, which equals $\mathbf{x}_*^{(i-1)} = f^{(i-1)}(...(f^{(1)}(\mathbf{x}_*))...)$. Then, for each $1 \leq j \leq n_i$:

$$\mathcal{D}(\mathbf{x}_*) \leftarrow \mathcal{D}(\mathbf{x}_*) \wedge \begin{cases} \mathbf{x}_j^{(i-1)} \leq 0 \wedge \mathbf{x}_j^{(i)} = \mathbf{x}_j^{(i-1)} & \text{if } (\mathbf{x}_*^{(i-1)})_j \leq 0 \\ \mathbf{x}_j^{(i-1)} \geq 0 \wedge \mathbf{x}_j^{(i)} = 0 & \text{if } (\mathbf{x}_*^{(i-1)})_j > 0. \end{cases}$$

**Iterative constraint solving.** We implement an optimization for solving LPs by lazily adding constraints as necessary. Given all constraints $\mathcal{C}$, we start off solving the LP with the subset of equality constraints $\hat{\mathcal{C}} \subseteq \mathcal{C}$, which yields a (possibly infeasible) solution $\mathbf{z}$. If $\mathbf{z}$ is feasible, then $\mathbf{z}$ is also an optimal solution to the original LP; otherwise, we add to $\hat{\mathcal{C}}$ the constraints in $\mathcal{C}$ that are not satisfied by $\mathbf{z}$ and repeat the process. This process always yields the correct solution, since in the worst case $\hat{\mathcal{C}}$ becomes equal to $\mathcal{C}$. In practice, this optimization is an order of magnitude faster than directly solving the LP with constraints $\mathcal{C}$.

**Single target label.** For simplicity, rather than minimize over $\rho(f, \mathbf{x}_*, \ell)$ for each $\ell \neq \ell_*$, we fix $\ell$ to be the second most probable label $\tilde{f}(\mathbf{x}_*)$; i.e.,

$$\hat{\rho}(f, \mathbf{x}_*) \overset{\text{def}}{=} \inf\{\epsilon \geq 0 \mid \hat{\mathcal{C}}_f(\mathbf{x}, \tilde{f}(\mathbf{x}_*)) \wedge \|\mathbf{x} - \mathbf{x}_*\|_\infty \leq \epsilon \text{ satisfiable}\}. \tag{5}$$

**Approximate robustness statistics.** We can use $\hat{\rho}$ in our statistics $\hat{\phi}$ and $\hat{\mu}$ defined in §2. Because $\hat{\rho}$ is an overapproximation of $\rho$ (i.e., $\hat{\rho}(f, \mathbf{x}_*) \geq \rho(f, \mathbf{x}_*)$), the estimates $\hat{\phi}$ and $\hat{\mu}$ may not be unbiased (in particular, $\hat{\phi}(f, \epsilon) \leq \phi(f, \epsilon)$). In §6, we show empirically that our algorithm produces substantially less biased estimates than existing algorithms for finding adversarial examples.

## 5  Improving Neural Net Robustness

**Finding adversarial examples.** We can use our algorithm for estimating $\hat{\rho}(f, \mathbf{x}_*)$ to compute adversarial examples. Given $\mathbf{x}_*$, the value of $\mathbf{x}$ computed by the optimization procedure used to solve (5) is an adversarial example for $\mathbf{x}_*$ with $\|\mathbf{x} - \mathbf{x}_*\|_\infty = \hat{\rho}(f, \mathbf{x}_*)$.

**Finetuning.** We use *fine-tuning* to reduce a neural net's susceptability to adversarial examples. First, we use an algorithm $\mathcal{A}$ to compute adversarial examples for each $\mathbf{x}_* \in X_{\text{train}}$ and add them to the training set. Then, we continue training the network on a the augmented training set at a reduced training rate. We can repeat this process multiple rounds (denoted $T$); at each round, we only consider $\mathbf{x}_*$ in the original training set (rather than the augmented training set).

| Neural Net | Accuracy (%) | Adversarial Frequency (%) | | Adversarial Severity (pixels) | |
|---|---|---|---|---|---|
| | | Baseline | Our Algo. | Baseline | Our Algo. |
| LeNet (Original) | 99.08 | 1.32 | 7.15 | 11.9 | 12.4 |
| Baseline ($T = 1$) | 99.14 | 1.02 | 6.89 | 11.0 | 12.3 |
| Baseline ($T = 2$) | 99.15 | 0.99 | 6.97 | 10.9 | 12.4 |
| Our Algo. ($T = 1$) | 99.17 | 1.18 | 5.40 | 12.8 | 12.2 |
| Our Algo. ($T = 2$) | 99.23 | 1.12 | 5.03 | 12.2 | 11.7 |

Table 1: Evaluation of fine-tuned networks. Our method discovers more adversarial examples than the baseline [21] for each neural net, hence producing better estimates. LeNet fine-tuned for $T = 1, 2$ rounds (bottom four rows) exhibit a notable increase in robustness compared to the original LeNet.

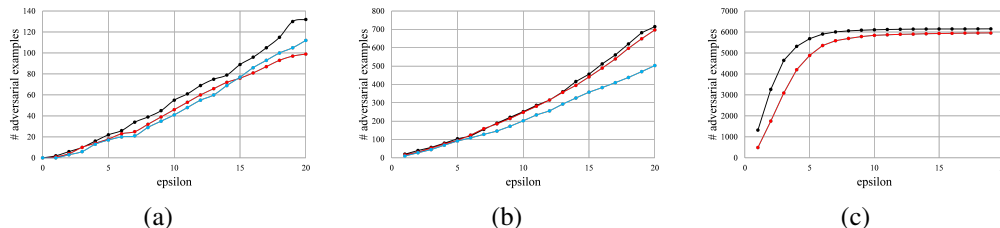

|   (a)   |   (b)   |   (c)   |

Figure 3: The cumulative number of test points $\mathbf{x}_*$ such that $\rho(f, \mathbf{x}_*) \leq \epsilon$ as a function of $\epsilon$. In (a) and (b), the neural nets are the original LeNet (black), LeNet fine-tuned with the baseline and $T = 2$ (red), and LeNet fine-tuned with our algorithm and $T = 2$ (blue); in (a), $\hat{\rho}$ is measured using the baseline, and in (b), $\hat{\rho}$ is measured using our algorithm. In (c), the neural nets are the original NiN (black) and NiN finetuned with our algorithm, and $\hat{\rho}$ is estimated using our algorithm.

**Rounding errors.**   MNIST images are represented as integers, so we must round the perturbation to obtain an image, which oftentimes results in non-adversarial examples. When fine-tuning, we add a constraint $\mathbf{x}_\ell^{(k)} \geq \mathbf{x}_{\ell'}^{(k)} + \alpha$ for all $\ell' \neq \ell$, which eliminates this problem by ensuring that the neural net has high confidence on its adversarial examples. In our experiments, we fix $\alpha = 3.0$.

Similarly, we modified the L-BFGS-B baseline so that during the line search over $c$, we only count $\mathbf{x}_* + \mathbf{r}$ as adversarial if $\mathbf{x}_\ell^{(k)} \geq \mathbf{x}_{\ell'}^{(k)} + \alpha$ for all $\ell' \neq \ell$. We choose $\alpha = 0.15$, since larger $\alpha$ causes the baseline to find significantly fewer adversarial examples, and small $\alpha$ results in smaller improvement in robustness. With this choice, rounding errors occur on 8.3% of the adversarial examples we find on the MNIST training set.

# 6   Experiments

## 6.1   Adversarial Images for CIFAR-10 and MNIST

We find adversarial examples for the neural net LeNet [12] (modified to use ReLUs instead of sigmoids) trained to classify MNIST [11], and for the network-in-network (NiN) neural net [13] trained to classify CIFAR-10 [9]. Both neural nets are trained using Caffe [8]. For MNIST, Figure 2 (b) shows an adversarial example (labeled 1) we find for the image in Figure 2 (a) labeled 3, and Figure 2 (c) shows the corresponding adversarial perturbation scaled so the difference is visible (it has $L_\infty$ norm 17). For CIFAR-10, Figure 2 (e) shows an adversarial example labeled "truck" for the image in Figure 2 (d) labeled "automobile", and Figure 2 (f) shows the corresponding scaled adversarial perturbation (which has $L_\infty$ norm 3).

## 6.2   Comparison to Other Algorithms on MNIST

We compare our algorithm for estimating $\rho$ to the baseline L-BFGS-B algorithm proposed by [21]. We use the tool provided by [22] to compute this baseline. For both algorithms, we use adversarial target label $\ell = \tilde{f}(\mathbf{x}_*)$. We use LeNet in our comparisons, since we find that it is substantially more robust than the neural nets considered in most previous work (including [21]). We also use versions

of LeNet fine-tuned using both our algorithm and the baseline with $T = 1, 2$. To focus on the most severe adversarial examples, we use a stricter threshold for robustness of $\epsilon = 20$ pixels.

We performed a similar comparison to the signed gradient algorithm proposed by [5] (with the signed gradient multiplied by $\epsilon = 20$ pixels). For LeNet, this algorithm found only one adversarial example on the MNIST test set (out of 10,000) and four adversarial examples on the MNIST training set (out of 60,000), so we omit results [2].

**Results.** In Figure 3, we plot the number of test points $\mathbf{x}_*$ for which $\hat{\rho}(f, \mathbf{x}_*) \leq \epsilon$, as a function of $\epsilon$, where $\hat{\rho}(f, \mathbf{x}_*)$ is estimated using (a) the baseline and (b) our algorithm. These plots compare the robustness of each neural network as a function of $\epsilon$. In Table 1, we show results evaluating the robustness of each neural net, including the adversarial frequency and the adversarial severity. The running time of our algorithm and the baseline algorithm are very similar; in both cases, computing $\hat{\rho}(f, \mathbf{x}_*)$ for a single input $\mathbf{x}_*$ takes about 1.5 seconds. For comparison, without our iterative constraint solving optimization, our algorithm took more than two minutes to run.

**Discussion.** For every neural net, our algorithm produces substantially higher estimates of the adversarial frequency. In other words, our algorithm estimates $\hat{\rho}(f, \mathbf{x}_*)$ with substantially better accuracy compared to the baseline.

According to the baseline metrics shown in Figure 3 (a), the baseline neural net (red) is similarly robust to our neural net (blue), and both are more robust than the original LeNet (black). Our neural net is actually more robust than the baseline neural net for smaller values of $\epsilon$, whereas the baseline neural net eventually becomes slightly more robust (i.e., where the red line dips below the blue line). This behavior is captured by our robustness statistics—the baseline neural net has lower adversarial frequency (so it has fewer adversarial examples with $\hat{\rho}(f, \mathbf{x}_*) \leq \epsilon$) but also has worse adversarial severity (since its adversarial examples are on average closer to the original points $\mathbf{x}_*$).

However, according to our metrics shown in Figure 3 (b), our neural net is substantially more robust than the baseline neural net. Again, this is reflected by our statistics—our neural net has substantially lower adversarial frequency compared to the baseline neural net, while maintaining similar adversarial severity. Taken together, our results suggest that the baseline neural net is overfitting to the adversarial examples found by the baseline algorithm. In particular, the baseline neural net does not learn the adversarial examples found by our algorithm. On the other hand, our neural net learns both the adversarial examples found by our algorithm *and* those found by the baseline algorithm.

### 6.3 Scaling to CIFAR-10

We also implemented our approach for the for the CIFAR-10 network-in-network (NiN) neural net [13], which obtains 91.31% test set accuracy. Computing $\hat{\rho}(f, \mathbf{x}_*)$ for a single input on NiN takes about 10-15 seconds on an 8-core CPU. Unlike LeNet, NiN suffers severely from adversarial examples—we measure a 61.5% adversarial frequency and an adversarial severity of 2.82 pixels. Our neural net (NiN fine-tuned using our algorithm and $T = 1$) has test set accuracy 90.35%, which is similar to the test set accuracy of the original NiN. As can be seen in Figure 3 (c), our neural net improves slightly in terms of robustness, especially for smaller $\epsilon$. As before, these improvements are reflected in our metrics—the adversarial frequency of our neural net drops slightly to 59.6%, and the adversarial severity improves to 3.88. Nevertheless, unlike LeNet, our fine-tuned version of NiN remains very prone to adversarial examples. In this case, we believe that new techniques are required to significantly improve robustness.

## 7 Conclusion

We have shown how to formulate, efficiently estimate, and improve the robustness of neural nets using an encoding of the robustness property as a constraint system. Future work includes devising better approaches to improving robustness on large neural nets such as NiN and studying properties beyond robustness.

## Footnotes

[1] Our neural net has 8 hidden units, but for this $\mathbf{x}_*$, 6 of the half-spaces entirely contain the convex region.

[2]Futhermore, the signed gradient algorithm cannot be used to estimate adversarial severity since all the adversarial examples it finds have $L_\infty$ norm $\epsilon$.

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
