[Reviews · NeurIPS 2016]

Reviewer 1

Summary

This paper provides a more formal definitions and metrics for measuring the robustness of neural networks to small perturbations of their inputs. The paper shows how to use these metrics to evaluate the robustness of deep networks for MNIST and CIFAR. The paper also shows how the proposed metrics can be approximated and encoded using a linear programming approach. The work presents some empirical evidence supporting the notion that prior approaches proposed to address the problem of robustness using adversarial examples can overfit the model to the adversarial examples. Importantly, the approach proposed here can also be used to improve robustness with respect to both previously proposed metrics and the metrics proposed here.

Qualitative Assessment

This paper presents a technically rigorous analysis of the issue of robustness to perturbations of inputs to neural networks with a focus on so called adversarial examples. The work examines a new formulation for defining and computing approximations to metrics consisting of the proposed definitions for: a) adversarial severity and b) adversarial frequency. As stated in the manuscript, the concept of adversarial frequency proposed here 'corresponds to the accuracy on adversarial examples used to measure robustness in [5, 19]', while 'Severity can be used to differentiate between neural nets with similar adversarial frequency'. I appreciate the technical rigour used to define these metrics, to formulate a linear programming approach to approximate them and the use of the approach to generate or find adversarial examples in a way that can be used to improve robustness. The key insights here revolve around restricting the analysis to the local linear regime of a neural network using ReLU units (and therefore simplifying the analysis). The manuscript should be more explicit when discussing the issue regarding the non-convexity of disjunctions (lines 197-212). As written it is not clear if the proposed strategy for dealing with disjunctions leads to another approximation or an exact (convex) reformulation. Please clarify in the author response using the exact language that would/could be added to the manuscript concerning this issue. The experimental work is reasonably well executed. If more significant practical impact could have been demonstrated on the CIFAR experiments then I would have likely give this work an even higher rating. Based on the valued added from the technical analysis, the LP convex optimization formulation and the experimental work I give this work a solid poster rating.

Confidence in this Review

2-Confident (read it all; understood it all reasonably well)


Reviewer 2

Summary

The authors propose two metrics for assessing the robustness of neural networks by an approximating algorithm for approximating a robustness criterion that was formerly introduced by Szegedy et al. Some experiments are run on the MNIST and the CIFAR-10 datasets to evaluate the performance of the proposed methods. The authors also explain how their method can be used to improve the robustness of neural networks.

Qualitative Assessment

The proposed approach is based on the nearest adversarial example (that is, an example with a different label) by L_infinity distance. It defines new metrics based on the fraction of examples within a particular distance and their average distance. These seem like reasonable measures, although the full CDF of adversarial distances is more informative. (They show these CDFs in the experiments, which is helpful.) Assuming that the space is completely linear is a very strong assumption that is not fully justified. Yes, the existence of adversarial examples has been attributed to the linear behavior of the network, but restricting the search to where the network is analytically linear could be problematic. A network may behave approximately linearly in a much larger region than where it is exactly linear. The most interesting result here is that they're able to find many adversarial examples that a previous baseline did not. Thus, while the restriction the linear region of a neural net seems restrictive, it still provides a better measurement and understanding of the vulnerabilities of these networks. Results on CIFAR-10 are informative, though mostly negative. After fine-tuning for accuracy, the robust network makes 10% more errors than before, and can still be fooled on ~60% of the examples by changing an average of 4 pixels (though that's an increase from 3). Minor issues: - Numeric citations would be improved by using author names as the noun instead of the reference number in phrases such as: "[6] takes the approach of [20]" - The paper title and abstract imply an approach that works on any neural network, but the method only applies to ReLUs. A revised title, or at least a revised abstract, seems appropriate. - The introduction needs to be revised. Specifically, it needs to explain more background from the references 5 and 20 in the beginning, such as why L_inf norm is the right choice for robustness. Also, rho is not defined before being referred to. The current order of the material makes the paper hard to follow. In the current format, the reader has to go back and forth and has to read those two references first to understand this paper. Typo: "when when" --> "when"

Confidence in this Review

2-Confident (read it all; understood it all reasonably well)


Reviewer 3

Summary

This paper discusses how to measure robustness of a neural net. The authors proposed a point-wise robustness metric, which is defined as the smallest perturbation (in \ell_{nifty} norm) that the prediction changes. Based on this metric, they further define two metrics, one is the probability (of the input) that the point wise robustness is below a threshold, and the other one is the conditional expected value of the point wise robustness when it is below a threshold. Using ReLU activation function, computing the robustness can be challenging. The authors propose a tractable convex relaxation that obtains an upper bound. Based on their proposed metric, the authors presented ways to improve robustness of neural nets, and demonstrated the superiority over existing schemes.

Qualitative Assessment

This paper discusses how to measure robustness of a neural net. The authors proposed a point-wise robustness metric, which is defined as the smallest perturbation (in \ell_{nifty} norm) that the prediction changes. Based on this metric, they further define two metrics, one is the probability (of the input) that the point wise robustness is below a threshold, and the other one is the conditional expected value of the point wise robustness when it is below a threshold. Using ReLU activation function, computing the robustness can be challenging. The authors propose a tractable convex relaxation that obtains an upper bound. Based on their proposed metric, the authors presented ways to improve robustness of neural nets, and demonstrated the superiority over existing schemes. I find this paper a piece of interesting contribution toward understanding neural networks. While the concept proposed is not exactly ground breaking, working out all the details does require non trivial effort, which leads to solid contribution to a very important and trendy topic. As such, I think it would benefit the community to have this paper in NIPS. Some more detailed comments: 1. The point wise robustness concept appears closely related to the concept of “margin”. I wonder whether the author can comment on this connection. 2. The concept of adversarial severity is less intuitive. Consider the following scenario: suppose we have improved the robustness of the network such that all points whose original point wise robustness is below \epsilon’ is now reduced to \epsilon (\epsilon’ > \epsilon). However, the adversarial severity for the more robust network is larger (implying being less robust), because of the condition expectation used. To address this, I propose to look at E[\rho \mathbb{1}(\rho\leq \epsilon)]. 3. L170: “when when” 4. I personally feel it may be more natural to have the subsection about the rounding error in the experiment section. 5. The authors mentioned that the proposed method does not improve robustness of NiN much, is there any intuitive explanation, even hypothetically?

Confidence in this Review

2-Confident (read it all; understood it all reasonably well)


Reviewer 4

Summary

This paper introduces an algorithm to generate adversarial examples for NNs (ie. images with an imperceptible perturbation that are mislabeled by the NN). This algorithm constrains the adversarial example to be in the linear region of the activations of the ReLUs of the NN. Experiments in MNIST show that for the NN called LeNet, the adversarial examples introduced in the paper seem to be more harming than the adversarial examples introduced in previous works (in terms of L_inf norm of the perturbation).

Qualitative Assessment

As in [5,20], this paper proposes a new algorithm to generate adverarial examples, and it uses the adversarial examples to re-train the NN. The algorithm to generate the adversarial examples is a bit different from previous works as it adds some additional constraints into to the optimization. I have two concerns regarding the motivation of these constraints: -Theoretical. The constraints that are added to the algorithm to generate the adversarial perturbation are motivated by: "Since adversarial examples exists because of the linearity in the neural net [5], we restrict our search to the region around the input in which the neural net is linear". Yet, adversarial examples have also been found outside these linear region around the input. A clear case of this is that when the adversarial perturbation is multiplied by a relativelly large constant, the adversarial examples still negativelly affects the NN [5]. Moreover, there are not any theoretical guarantees that by constraining the adversarial examples to be in the aforementioned linear region, the adversarial example has minimum perturbation. There could be adversarial examples with smaller norm outside the search space of the algorithm introduced in the paper. -Experimental. The conclusions extracted from the results do not seem to be general. The NN architectures are not state-of-the-art and not standard, and are applied in very simple datasets (MNIST and CIFAR). There are no sanity checks that the baseline of [20] has been correctly reproduced. Also, results show marginal improvements in MNIST over the baseline, and in CIFAR, the comparision with the baseline has been omitted. Other comments: -The results in CIFAR show that this method may not have much applicability to reduce the adversarial examples, as it takes about 15 seconds to generate one adversarial example using 8 CPUs. This will make it difficult to scale to ImageNet, and create sufficient adversarial examples to re-train the NN. -The first time that $x_\star$ is introduced, it has been never defined before. -It has been assumed that L_inf norm is the best measure to analyze the perceptibility of the adversarial perturbation. -It has been mentioned that "[3] seeks to explain why neural nets may generalize well despite poor robustness properties". A more detailed explanation of [3] seems necessary here. Maybe the motivation of the constraints could be done from [3].

Confidence in this Review

3-Expert (read the paper in detail, know the area, quite certain of my opinion)


Reviewer 5

Summary

The paper suggests to use linear programing in order to find adversarial examples for ReLu networks and defines measures of robustness. The paper also distinguishes the method that generates the adversarial examples from the method that measures robustness.

Qualitative Assessment

Given the current interest in robustness properties of deep neural networks, this is definitely an interesting topic to investigate. The paper is well written and easy to follow. The results are convincing and I think that researchers from the deep learning community would benefit from reading this paper. However, I have two main points of criticism: First, while using linear programing to find adversarial examples is appealing, the approach presented in the paper relies on linear properties of ReLu networks and therefore hampers its usage. Second, the methods proposed in this paper seems to be superior over previous approaches at identifying adversarial examples, they do not improve the performance on the test set significantly. The authors also fail to demonstrate the superiority of their method in other data sets beside Mnist. Smaller remarks: I find the formalization of the linear constraints a bit exhausting. It is clear that at a given point x only one of the disjunction constrains is needed. I therefore don’t see the point in introducing disjunctions only to later disregard them.

Confidence in this Review

3-Expert (read the paper in detail, know the area, quite certain of my opinion)


Reviewer 6

Summary

The paper suggests a new way of constructing adversarial examples for NN that are not necessarily at the direction of the signed gradient (of wrong labels). The idea is to encode the NN as a set of constraints and solve a LP problem whose (feasible) solution provides an adversarial example as well as the robustness measure. The authors also formalize the notion of robustness and propose 2 new statistics for it.

Qualitative Assessment

1) equation between lines 212-123: i think the conditions are switched. 2) The authors argue that current robustness estimations are biased because they are measured on data produced by the inspected algorithm itself. However what they propose is not much different. They still use their algorithm to measure rho. 3) They claim is that they are less biased just because they find more bad examples. Please support this claim. 4) I believe the proposed algorithm is indeed biased because in the process of removing disjunction constraints the algorithm always chooses to remove the one not satisfied by the seed. Can you say anything about a stochastic version to choose between disjunctions? 5) Overall the paper is hard to follow. Symbols and equations are not formulated enough. For example line 147: what's the dimensionality of x*? what does (.)_i relate to? rows? columns? 6) line 64: i believe it should be "x* with true label l*"

Confidence in this Review

2-Confident (read it all; understood it all reasonably well)